# Mast Cells in Allergic and Non-Allergic Upper Airways Diseases: Sentinel in the Watchtower

**DOI:** 10.3390/ijms252312615

**Published:** 2024-11-24

**Authors:** Giovanni Costanzo, Marta Marchetti, Andrea Giovanni Ledda, Giada Sambugaro, Martina Bullita, Giovanni Paoletti, Enrico Heffler, Davide Firinu, Giulia Anna Maria Luigia Costanzo

**Affiliations:** 1Personalized Medicine, Asthma and Allergy, IRCCS Humanitas Research Hospital, 20089 Rozzano, Italy; marta.marchetti@humanitas.it (M.M.); giovanni.paoletti@hunimed.eu (G.P.); enrico.heffler@hunimed.eu (E.H.); 2Department of Medical Sciences and Public Health, University of Cagliari, 09124 Monserrato, Italy; andrea.giovanni.ledda@gmail.com (A.G.L.); g.sambugaro@studenti.unica.it (G.S.); davide.firinu@unica.it (D.F.); giuliaam.costanzo@unica.it (G.A.M.L.C.); 3Department of Biomedical Sciences, Humanitas University, 20072 Pieve Emanuele, Italy

**Keywords:** mast cells, allergy, rhinitis, non-allergic rhinitis, rhinosinusitis, nasal polyps

## Abstract

Mast cells are immune system cells with the most disparate functions, but are also among the least understood. Mast cells are implicated in several known pathological processes, tissue homeostasis, and wound repair. However, they owe their notoriety to allergic diseases, of which they represent the effector cell par excellence. In both allergic and not upper airway pathologies, mast cells play a key role. Exploring the mechanisms through which these cells carry out their physiological and pathological function may help us give a new perspective on existing therapies and identify new ones. A focus will be placed on non-allergic rhinitis, a poorly recognized and often neglected condition with complex management, where the role of the mast cell is crucial in the pathogenetic, clinical, and prognostic aspects.

## 1. Introduction

Mast cells (MCs) were first described by Paul Ehrlich in 1877 [1]. These cells have a very early embryonic origin and develop complexly through various stages of maturation. MCs play a fundamental role in the immune system, involving numerous functions ranging from defense against parasites [2] and bacteria [3] to the induction of inflammation and more complex roles such as immunomodulation and tissue repair [4,5]. In adults, MCs originate in the bone marrow from multipotent hematopoietic stem cells through a granulocyte/monocyte lineage, CD34+ and CD117+ [6]. Immediately after conception, the MCs increase in number and complexity. Fetal immature hypogranular MCs derived from stem cells can be found starting from the early stages of embryonic development in the yolk sac and the fetal liver [7]. Subsequently, it is possible to observe CD45+ CD117+ cells in fetal skin and the airways [8]. In adults, the MCs are primarily found in peripheral tissues, released as undifferentiated mononuclear cells through “homing” mechanisms directly from the bloodstream [9]. Immature forms of MCs (immature progenitors of human mast cells, MCps) can be detected in adults in both peripheral blood and bone marrow [4,10]. The tyrosine-protein kinase KIT (KIT in short) receptor or mast/stem cell growth factor receptor (SCFR) receptor regulates the MCs’ development, maturation, and survival. Treatments with imatinib, a KIT signaling inhibitor, induce a numerical reduction in mature MCs but do not affect MCps [11].

Numerous studies have proven the presence of a variety of MC subclasses in different organs and within the same organ, with important differences in maturation, granule content, and expression in receptors [12,13]. Nevertheless, a classical division based on their pattern of protease expression is still useful and used widely: (1) mast cells that produce both tryptase and chymase (MCTC), and (2) mast cells that produce only tryptase (MCT). MCT cells primarily serve a defensive function, are located on mucosal surfaces, and interact with the immune system [14]. MCTCs are more involved in tissue repair, fibrotic reactions, and angiogenesis [15].

Mast cells are the human cells with the highest number of receptors [16]. The most well-known receptor is the IgE receptor, whose activation leads to allergic response. MCs possess a vast repertoire of non-IgE receptors that allow mast cells to adapt and meet the various functional requirements of the host tissue and respond to a multitude of stimuli (tryptase, complement bacterial products, neuropeptides, platelet-activating factor, hyperosmolarity, and stress) [16,17]. Among the non-IgE receptors, we can distinguish FcγRI receptors that allow mast cells to bind to IgG, Toll-like receptors, C5a receptors, pattern recognition receptors, nuclear receptors, receptors for alarmins, integrins, neuropeptides such as substance P, nerve growth factor (NGF), calcitonin gene-related peptide (CGRP), and vasoactive intestinal peptide (VIP), in addition to vitamin D [18]. One of the latest receptors recently described is the Mas-related G protein-coupled receptor X2 (MRGPRX2) [19], which is highly expressed in human skin mast cells (but absent in the airways) and can be activated by a wide range of cationic molecules, neuropeptides, host defense peptides, and various drugs; it is considered a key actor in the so-called pseudoallergic reactions [20] and a promising potential therapeutic target [21]. The spleen tyrosine kinase (SYK) and Bruton’s tyrosine kinase (BTK) are the most well-known and studied signaling proteins of mast cells, following the activation of the high-affinity IgE receptor (FcεRI), resulting in the release of calcium from stores in the endoplasmic reticulum and phosphorylation of targets, leading to the subsequent production and release of mediators. Other signal transducers with potential therapeutic potential include calcineurin [22], extracellular signal-regulated kinase (ERK) [23], and hypoxia-inducible factor 1-α (HIF-1α) [24].

Mast cells release a wide range of preformed mediators and signaling molecules into the surrounding microenvironment, influencing various tissue-resident cells, such as fibroblasts, smooth muscle cells, endothelial cells, and epithelial cells [4]. Furthermore, they synthesize and release both serines and metalloproteases, which cause the degradation of the extracellular matrix and tissue remodeling [25]. Histamine is the main mediator released from the degranulation of mast cells [26]. The release of histamine leads to the dilation of post-capillary venules, the activation of the endothelium, and an increase in the permeability of blood vessels, resulting in very different signs and symptoms depending on the affected tissue and organ (e.g., swellings typical of hives or nasal obstruction due to mucosal edema in allergic rhinitis) [27]. On the other hand, the chronic release of tissue repair cytokines by mast cells can lead to detrimental effects for the host, such as fibrosis in asthma [28]. Among all, serum tryptase is used as a biomarker for mast cell degranulation. A sharp increase in tryptase levels indicates systemic hypersensitivity or anaphylaxis, while chronically elevated tryptase levels should raise suspicion of mastocytosis or hereditary α-tryptasemia [4,29,30,31,32,33].

MCs are highly granular tissue-resident cells, part of the innate immune and neuroimmune systems, and the leading players in allergic reactions, anaphylaxis, and many other diseases. MCs are distributed throughout the body, but they are mainly present near barriers such as the skin and the mucous membranes of the lungs, digestive tract, conjunctiva, and nose [5,29]. In the immune system cell army, MCs act as the sentinels, stationed at the borders to sense the surrounding environment. They can quickly respond to external stimuli and start a well-organized process of inflammation and subsequent tissue repair [34]. Putting aside its role in non-allergic diseases such as systemic mastocytosis, MC amplification is observed in various disorders linked to type 2 (T2) inflammation, such as food allergy, eosinophilic esophagitis (EoE), and atopic dermatitis (AD), but are most studied in airway disease, including rhinitis [35] and asthma [28]. MCs contribute to the development and severity of T2 inflammatory diseases by engaging in eicosanoid biosynthesis, cytokine generation, and the discharge of pre-existing mediators such as histamine and proteases [36]. However, it has been demonstrated that, at least in asthma, MCs contribute to the pathogenesis of the disease regardless of the T2 high or low phenotype, as mast cell tryptase is elevated in severe asthma patients independent of type 2 biomarker status [37].

Nevertheless, MCs also have a significant and frequently overlooked function in maintaining tissue balance, addressing infections, and promoting wound healing, neurogenesis, and vasculogenesis [5,38,39]. The release of potent mediators and cytokines has an evolutionary role in host defense, which ultimately may provoke or exacerbate pathologies [28]. Therefore, MCs should be regarded as a double-edged sword.

In this review, we will describe the role of mast cells in the main allergic and non-allergic upper airway diseases. Proposing an in-depth description of the biological functions of mast cells is beyond the scope of this article. Therefore, only those aspects useful for understanding the pathogenesis of the diseases discussed below have been summarized. In the following sections, allergic rhinitis, chronic rhinosinusitis with nasal polyposis and non-allergic rhinitis will be addressed and analyzed in their general aspects and in such a way as to highlight aspects in common, with particular attention to the role of mast cells. In particular, non-allergic rhinitis will be addressed in its clinical and cytological features as we believe it is an often-overlooked topic with important research potential. Finally, in light of the knowledge that has emerged on the functioning of the pathologies described, a section will be dedicated to management, with a particular focus on current and future therapeutic strategies that target mast cells.

## 2. Mast Cells in Upper Airways

### 2.1. Allergic Rhinitis

Rhinitis is defined as inflammation of the lining of the nose. Allergic rhinitis (AR), like other allergic conditions, occurs when the immune system of a sensitized person overreacts to harmless substances called allergens [40].

AR is the most prevalent form of non-infectious rhinitis [40,41]. In industrialized nations, the prevalence of rhinitis in the general population ranges from 10 to 40%. AR affects 3.4% of children at 4 years of age and 27.3% at 18 years of age, while the prevalence rate in adults has been estimated at 29.8%. The risk factors for AR include a family-related predisposition, male sex, being born during the pollen season, being the firstborn, early-life antibiotic administration, maternal smoking, exposure to indoor allergens and levels of IgE in the blood > 100 IU/mL before the age of 6 together with the presence of allergen-specific IgE [42].

From a pathogenetic point of view, exposure to inhaled allergens in sensitized individuals initiates a cascade of events leading to allergic reaction symptoms. This process involves binding the allergen to a specific IgE antibody located on the surface of MCs, which then triggers the degranulation and the release of mediators. This event supports the initial allergic reaction and the subsequent recruitment and infiltration of effector cells, such as B cells, T cells, and eosinophils, to the site of inflammation [40]. In particular, it was found that histamine produced from MCs’ degranulation initiate the loss of the epithelial barrier during the early stage of the allergic immune response, which is then maintained by Th2 cells and MCs themself, through the production of interleukin-4 (IL-4) and IL-13 [43].

The typical clinical presentation is sneezing, excessive nasal discharge, obstruction, and itching. These symptoms are often accompanied by cough, postnasal drip, irritability, fatigue, and itching of the palate and inner ear. Ocular symptoms, including pruritus, lacrimation, and ocular burning sensation, could be present [42,44,45]. Traditionally, AR has been categorized as either seasonal or perennial, depending on symptoms and the associated allergens. However, this classification is not universally applicable, particularly in countries where pollinosis occurs yearly. As a result, an alternative approach is to classify AR based on the severity (mild, moderate, severe) and duration (intermittent, persistent) of symptoms [46]. Due to the overlapping symptoms of AR and non-allergic rhinitis (NAR), it is not possible to make a differential diagnosis based on the clinical presentation, so it is mandatory to employ diagnostic methods like a skin prick test (SPT) or in vitro IgE test to document the allergic sensitization [45]. In some cases, nasal endoscopy, computed tomography (CT) scan, and nasal cytology [47] might be useful for deeper disease characterization.

#### Role of the Mast Cells in Pathophysiology of Allergic Rhinitis

In AR, MCs are the main cells responsible for the symptoms and a key element in every stage of the allergy. Indeed, in the early stage, MCs are the first cells to be activated and triggered, determining the immediate reaction. In the late inflammatory phase, they organize the cellular recruitment that perpetuates clinical chronicity.

The nasal mucosa serves as the initial defense mechanism of the respiratory system coming into contact with different pathogens [36]. When stimulated by allergens in sensitized patients, nasal MCs migrate to the surface of the nasal mucosa, where they undergo degranulation [48], which is crucial in the pathogenesis of AR [49] as degranulating MCs release many substances such as histamine, proteases, prostaglandins (PGs), cysteinyl leukotrienes (CysLTs), and type 2 cytokines, including IL-5 and IL-13 [50]. Research on the secretion of histamine, prostaglandin D2 (PGD2), and tryptase during the immediate nasal response confirmed the crucial role of MCs in developing immediate symptoms in AR [51]. MCs also coordinate the multiple components of the immune response associated with IgE, such as the late-phase allergic response and chronic allergic inflammation through the release of mediators and orchestrate the activity of other cells by direct cell–cell interaction [52]. Furthermore, the generated IgE can augment the expression of high-affinity IgE receptor, (FcεRI) in nasal MCs (NMCs) and the subsequent discharge of mediators and may facilitate the binding of a greater number of IgE-antigen complexes, hence augmenting the production of inflammatory mediators through allergen- and IgE-mediated processes, creating a critical process that amplifies positive feedback [53] [Figure 1].

### 2.2. Non-Allergic Rhinitis (NAR)

While the borders of the pathology are clear and transversely accepted for allergic forms, different definitions can be found in the literature for NAR, leading to the use of imprecise terminology. Definitions such as non-allergic and non-infectious rhinitis [54], intrinsic or idiopathic rhinitis [55], vasomotor rhinitis [56], and non-allergic vasomotor rhinitis can be widely found in the literature used as synonyms [57].

In 2017, the European Academy of Allergy and Clinical Immunology (EAACI) addressed the topic. It published a position paper [54] in which chronic rhinitis is divided according to the pathogenesis into infectious (usually caused by a virus, resulting as a common cold), allergic (the most common, induced by allergen inhalation by sensitized individuals [58]) and nonallergic noninfectious rhinitis (NANIR, NAR in short), with mixed forms [59,60]. The same authors collect a variety of established pathologies under the term NAR, such as drug-induced rhinitis, rhinitis of the elderly, hormonal rhinitis, nonallergic occupational rhinitis, gustatory rhinitis, and idiopathic rhinitis.

In general, the term NAR indicates a form of rhinitis in which it has not been possible to identify the underlying cause after excluding the two most probable and common etiologies, i.e., allergic and infectious [61]. However, infectious rhinitis is usually acute or, at most, subacute. In contrast, allergic rhinitis is chronic although it can present a seasonal or recurrent pattern depending on whether or not the trigger allergen is present [44]. Therefore, NAR usually refers to a form of chronic rhinitis, as opposed to the other complementary form of chronic rhinitis, the allergic one. There is no consensus about how long symptoms should be present to establish chronicity. The lack of a unifying pathogenetic theory makes it difficult to find a unanimous consensus. Moreover, the term NAR gathers a set of different pathologies that still need to be completely understood and defined.

Rhinitis affects 10 to 40 percent of the population in industrialized countries [41,45,61,62] and according to the Joint Task Force on Practice Parameters in Allergy, Asthma, and Immunology, NAR may be responsible for more than 50 percent of rhinitis, which may have the nonallergic form (alone or as a mixed disease) [63].

As just described, by definition, NAR is a diagnosis of exclusion. Consequently, the patient may be assigned the NAR label in some cases because the correct allergen sensitization has not been identified. This is the case with local allergic rhinitis (LAR) or “entopy”, a localized nasal allergic response in patients with negative SPT and absence of detectable specific IgE (sIgE) to inhalant allergens in the blood [64]. These patients develop a local-only sensitization and reaction, with nasal sIgE [65] and a positive nasal allergen provocation test (NAPT) response [66]. The symptoms are AR-like, elicited by natural exposure to aeroallergens, and the therapy with nasal corticosteroids is generally effective. LAR is, to date, a debated and complex entity, whose diagnosis is not standardized by international guidelines and is based on a combination of NAPT, assessment of local IgE, basophil activation test (BAT), and nasal cytology [67]. It is ultimately a form of AR [68].

However, the exclusionary definition of NAR should not lead to the belief that the term was created solely to fill a gap in the taxonomy of rhinitis. The inflammatory cytokine pattern in chronic NAR is different from controls and patients with AR: studies showed that NAR and AR patients have similar serum levels of IL-16 and IL-17 compared to the control group [69], but the NAR has a higher level of costimulatory molecule sCD48 with a significant correlation between the serum level of sCD48 and the number and percentage of eosinophils and eosinophil cationic protein [70]. Eotaxin-1/CCL11 serum levels were also found significantly higher in the non-allergic group [70] and the TRPV1/SP pathway is upregulated, leading to neurogenic inflammation [71].

Moreover, many clinical differences have been found: NAR presents a general female predominance and occurs later in life [72], even though many classifications include elderly rhinitis. The symptoms are more likely to be perennial than seasonal [63] and, unlike AR, the most frequent and prominent clinical manifestations are nasal blockage and postnasal drip [62].

This suggests that these patients are not merely allergy sufferers who are falsely negative in allergy tests.

#### Role of the Mast Cells in the Pathophysiology of Non-Allergic Rhinitis

Pathology and cytology have come to help untangle the classification question, putting aside the etiopathogenetic approach and placing the cell at the center [73]. Few papers in the literature compare cytologic aspects of AR e NAR.

In 1997, Berger et al. [74] studied nasal full-thickened mucosa and compared the number of MCs in the inferior turbinate of patients with AR and NAR with normal controls, finally concluding that the density of MCs was similar in AR and NAR and significantly higher than in normal controls. The same results were found in a more recent study in 2001 by Powe et al. [75], demonstrating similar epithelial patterns in mast cells, eosinophils, and IgE+ cells, not exclusively mast cells (i.e., plasma cells and macrophages), with the latter more increased in the allergic group. The authors concluded that a localized IgE-mediated Th2 inflammatory mechanism involving mast cells in NAR may occur, unlike the systemic response in AR. This model was further developed in the aforementioned “entopy” [64], which became a paradigm of local allergy [76].

A cell-based well-established nasal disease is NARES. The term NARES (NAR with eosinophilia syndrome) was first used in 1981 by Jacobs et al. [77] to describe chronic perennial rhinitis with greater than 20% of the cells in nasal smears being eosinophils in the absence of sIgE in skin and blood testing. The pathophysiology of NARES is still poorly understood, but a key component involves self-perpetuating eosinophilic nasal inflammation, which can eventually lead to polyposis [78].

Subsequently, the rapid diffusion of nasal cytology allowed further studies in this regard, identifying new pathological entities: NARNE (NAR with neutrophils) NARMA (NAR with mast cell), and NARESMA (NAR with eosinophils and mast cell) [79]. Nasal cytology is a simple, cheap, point-of-care, non-invasive, and rapid technique that allows an outpatient setting to acquire information through cell counting and observation. This tool makes it possible to investigate the immunological response in vivo in the patient [80]. Although it can guide the diagnosis of allergic and infectious rhinitis, it is particularly important in characterizing otherwise neglected forms of NAR according to the predominant cell type [81]. Growing evidence is being collected that it is a reliable tool that might be used in association with other techniques for better defying the rhinitis endotype with crucial prognostic repercussions on the patient [82,83].

NARNE is characterized by a predominant infiltration of neutrophils (>20%) in the absence of bacterial or spores/fungal hyphae; NARMA is defined by the presence of MCs > 10%, partially degranulated, and can be considered a transitional form leading to NARESMA, a novel entity with the presence of eosinophils and MCs in variable proportion with relevant degranulation [81]. There is a lack of epidemiological data about cellular NAR forms, but in an unselected group of Italian NAR patients, the distribution was NARES 30%, NARESMA 28%, NARMA 22%, and NARNE 20% [79].

Following the definition of the aforementioned pathological entities, clinical and prognostic correlates were progressively identified [80]. In particular, the presence of MCs in nasal cytology indicates a more severe condition and does not represent mere descriptive data, but an important element characterizing NAR.

The clinical presentation of NARMA and NARESMA is usually the most severe above all the NAR. In a large study of its kind conducted in 2007 on 176 NAR subjects, patients with NARESMA had the worst score in the quality-of-life questionnaire, especially in the sleep and social dimension, and presented a more intense impairment for dry nose, snoring, sore throat, nasal obstruction, and sore throat, with more common snoring and nocturnal awakening. Moreover, this clinical aspect is confirmed by functional findings: the nasal resistance measured with rhinomanometry was higher in NARESMA than in the others. Finally, it was more often associated with the presence of asthma and/or nasal sinus polyposis, with more frequent relapses. Finally, patients with NARESMA have a higher risk of undergoing turbinate surgery [79].

In conclusion, regardless of the boundaries of the definition and the difficulties in making a diagnosis, the presence of mast cells represents a crucial element indicating a more serious disease regardless of the etiology. Increasing data suggest that MCs also play a role in infectious rhinitis, acting as innate immune cells against various pathogens and initiating defensive responses [84] [Figure 2].

### 2.3. Chronic Rhinosinusitis with Nasal Polyps

Chronic rhinosinusitis with nasal polyps (CRSwNP) is an inflammatory disease of the upper airways that significantly impacts affected patients’ health and quality of life. It has been estimated that the prevalence of this disease is about 2-4% in the general population [85,86].

The pathogenesis of CRSwNP is characterized by an inflammatory disorder, mostly associated with type 2 inflammation. This is characterized by epithelial barrier dysfunction causing activation of Th2 cells and innate lymphoid cell 2 (ILC-2) and release of IL-4, IL-5, and IL-13 [87]. As a result, eosinophils are recruited to the tissue level, IgE production is increased, and structural changes in the nasal mucosa develop with the formation of polyps [88]. Histomorphologic characterization of CRSwNP reveals frequent epithelial damage, a thickened basement membrane, and mostly edematous to sometimes fibrotic stromal tissue, with a reduced number of vessels and glands but virtually no neuronal structures [89].

By definition, symptoms associated with CRSwNP include loss of sense of smell (anosmia), nasal obstruction or blockage, anterior or posterior nasal drainage, and facial pressure for more than 12 weeks [90]. In addition, there must be objective evidence of sinonasal inflammation and nasal polyps on sinus CT scans and/or nasal endoscopy [91]. The disorder is associated with significant consumption of medical resources and reduced quality of life, especially during acute exacerbations. Several comorbid conditions, such as allergic rhinitis, asthma, sleep disorders, and gastroesophageal reflux disease, are commonly reported in patients with CRSwNP [92].

#### Role of the Mast Cells in Pathophysiology of Chronic Rhinosinusitis with Nasal Polyps

A plethora of factors contribute to the pathogenesis of CRSwNP.

Defects in innate airway epithelial barrier function, including reduced expression of antimicrobial products and loss of barrier integrity, combined with colonization by *fungi* and *bacteria*, likely play a critical role in developing chronic inflammation in CRSwNP. This chronic inflammation is characterized by elevated expression of key inflammatory cytokines and chemokines, including IL-5, thymic stromal lymphopoietin, and CCL11. These contribute to initiating and perpetuating this chronic inflammatory response. These factors likely synergize to drive the influx of various immune cells, including eosinophils, MCs, group 2 innate lymphoid cells, and lymphocytes, participating in the chronic inflammatory response in nasal polyps [93].

The potential significance of MCs in chronic rhinosinusitis (CRS) pathogenesis has been considered due to their production of various cytokines that activate eosinophils, molecules that promote tissue remodeling, and chemical mediators that can cause tissue edema. Distinct MC subsets infiltrate the airway mucosa in T2 disease, including subepithelial MCs expressing the MCTC and MCT [94,95].

Studies have reported a significant increase in mast cells in the epithelium and glands of nasal polyps (NPs) compared with controls and patients with CRS without NPs. MCs in the epithelium of NPs show upregulation of tryptase and CPA3, while those in the glands express all three proteases typical of the MCTC phenotype (tryptase, carboxypeptidase, and chymase). This suggests that different populations of MCs in the mucosal epithelium and submucosal glands may play a crucial role in developing NPs [94].

In addition, the expression of T-cell/transmembrane immunoglobulin and mucin domain protein 3 (TIM-3), a receptor that promotes MC activation and production of cytokines, is increased in MCs infiltrating nasal polyps [96]. Moreover, the expression of tryptase mRNA was significantly higher in NPs from CRS patients compared to controls, with increased tryptase-positive mast cells in various regions of NPs, especially in eosinophilic CRS (ECRS) polyps. IgE-positive mast cells were more abundant in ECRS polyps compared to non-ECRS polyps. These results suggest that IgE-mediated mast cell activation may contribute to the pathogenesis of ECRS as well [97].

In conclusion, MCs have a crucial role in the pathogenesis of CRSwNP, orchestrating eosinophilic inflammation and causing the most severe forms. Moreover, the presence of MCs in the nasal cytology of patients affected by CRSwNP constitutes an independent negative prognostic factor as it is associated with a higher risk of recurrence after surgery [98], refractory to treatment [99], and overall severity [100,101].

## 3. Treatment of Rhinopaties Focusing on Mast-Cell Centered Therapies

After discussing the role of mast cells in various rhinopaties, it is easy to demonstrate how therapies already widely used and consolidated often act on these cells to alleviate symptoms and reduce inflammation.

Due to its low cost, great availability, and virtual absence of adverse effects, nasal saline irrigation is the first recommendation for alleviating rhinitis symptoms. Until recently, its effectiveness was anecdotal and dictated by a well-established common practice, when it was confirmed by a trial and meta-analysis in all forms of rhinitis [102].

In AR, once the trigger allergen has been found, allergen avoidance should be the first step, also with high-efficiency particulate air filters, barrier measures such as non-pharmacological intranasal cellulose powder, and lipid microemulsion [103].

As advocated by the 2016 Allergic Rhinitis and its Impact on Asthma (ARIA) guidelines [44], in patients with AR, the use of intranasal corticosteroid (INCS) is suggested as first-line therapy with or without oral antihistamines or intranasal antihistamines depending on the clinical presentation and severity. INCS reduces the mucosal inflammation that underlies the signs and symptoms of the disease [104], leading to modifications in gene transcription and a reduction in the synthesis of cytokines, which ultimately attenuates the recruitment, survival, and activity of inflammatory immune cells, like MCs [30]. Antihistamines, both oral and local, limit the effect of histamine, the main mediator produced by MCs and the principal responsible for most symptoms such as nose itching, rhinorrhea, sneezing, and nasal obstruction. In the same guidelines, anti-leukotrienes, also widely produced by mast cells, are recommended in selected cases [4]. The use of topic decongestants, anticholinergics, and chromones is more limited [30,45].

Cromolyn sodium functions as an MC stabilizer, inhibiting the release of inflammatory mediators [105] and serving as an immunomodulator, thereby reducing allergic inflammation [106]. The efficacy of intranasal cromolyn is limited, but shows an excellent safety record, also in pregnancy [45].

Leukotriene receptor antagonists (LTRAs) act on leukotriene receptors on immune cells blocking the leukotriene-mediated inflammatory cascade. Leukotrienes are secreted from activated MCs and other inflammatory cells such as eosinophils. LTRAs were found to provide significant improvements in nasal symptoms and the quality of life attenuating nasal obstruction and rhinorrhea [107]. Nevertheless, LTRAs are generally not recommended for the initial treatment of rhinitis presenting a higher risk profile than other alternative and preferred therapies, with reported severe neuropsychiatric events [45].

Allergen immunotherapy (AIT) is currently the only available disease-modifying therapeutic option. It is usually administered in increasing amounts at regular intervals for a long time (three to five years) to modulate the patient’s immune system’s response to the trigger allergen [108,109]. AIT acts directly and indirectly on MCs, leading to desensitization, generation of regulatory lymphocyte responses, and regulation of IgE production, which determines a reduction in MCs and eosinophil activity [110,111].

New drugs are constantly developing [103], focusing on novel molecules tested in mouse models [112,113,114] and recently found and orphan receptors as potential therapeutic targets [115]. Also, old drugs may potentially find a new life: local cyclosporine, an immunosuppressive molecule that inhibits calcineurin in Th cells, has been tested in mice with ovalbumin-induced AR with similar results to INCS and no side effect was observed [116].

Analyzing the therapy of CRSwNP, various steps rely more on intranasal drugs, especially corticosteroids. The first treatment step for all patients includes medical management with INCS, with an increase in the intensity determined by the severity of the disease [117].

If there is a complete obstruction in the nasal cavity, a short course of oral glucocorticoids could improve access to the INCS. In patients not adequately responsive to medical therapies, surgical intervention (endoscopic sinus surgery, ESS) is often required, with a conservative approach (removal of all nasal polyps while preserving the sinus mucosa) or, if indicated, a more radical one such as reboot surgery (complete removal of polyps and sinus mucosa) [118,119]. However, about 35% of CRSwNP patients undergoing surgery tend to relapse as early as the first six months post-surgery, with no real resolution of symptoms nor improvement in quality of life [120].

The EPOS/EUFOREA 2023 update on the indication and evaluation of biologics in CRSwNP recommends considering biologics in uncontrolled patients despite appropriate medical and surgical therapies [121].

Among the biological therapies approved for CRSwNP treatment, those that act the most on MCs are omalizumab [122] and dupilumab [123]. Dupilumab blocks the IL-4Rα subunit of IL-4 and IL-13 receptors, thus inhibiting IgE production and indirectly limiting MCs’ activation and degranulation [40]. Omalizumab interferes with the bond of IgE and its receptors on MCs, suppressing their activation. Among the wide and complex immunological pathways in which those two drugs operate, the interference on the mast cells must be considered one of the reasons for the clinical efficacy [40]. Moreover, although IL-5 exerts its main function on eosinophils, MCs possess receptors for the cytokine and this might indicate that therapies directed against the IL-5 pathway approved in CRSwNP (e.g., mepolizumab) might relevantly influence MCs’ functions [124].

In the wide and heterogeneous scenario of rhinitis, NAR is surely the least studied and understood, and the therapeutic approach is no exception. Few studies have been found comparing the efficacy of rhinitis therapies in AR and NAR. Based on the current literature, a 2020 comprehensive consensus on rhinopaties [45] suggests either intranasal antihistamines or INCS as first-line monotherapy for NAR and intranasal ipratropium if rhinorrhea is the main nasal symptom, non-recommendingthe use of an oral leukotriene receptor antagonist. The role and frequent presence of MCs in NAR explain the effectiveness of this approach, even if quantifying the efficacy in NAR is more challenging, mostly because of the vague definition. A 2015 study by Kirtsreesakul et al. [125] compared the response to INCS in AR and NAR, and although both had good steroid responses, the patients with NAR showed less improvement than those with AR.

A recent meta-analysis has questioned the efficacy of INCS in NAR. However, it must be noted that the data were collected from a group of non-homogeneous patients across various studies [126].

## 4. Discussion

Mast cells are an integral part of the immune system, bridging innate and adaptive immunity, yet they are often underestimated. Over the years, new and unexpected functions have been discovered, far beyond the role in allergic diseases for which they are mainly known.

The pathological function goes hand in hand with the physiological one of maintaining and restoring homeostasis. More and more studies demonstrate how MCs can play a role even in the context of diseases distant from allergies. For instance, in cancer, MCs can both fight the tumor and promote the development of a cancerogenic inflammatory environment [127,128].

The role highlighted in chronic pathologies sparks the debate on the label often given to the MCs. Studies on chronic immune-mediated diseases such as systemic sclerosis demonstrate how they contribute to fibrosis [129]. In asthma, it has been shown how MCs affect the remodeling of the lower airways, which are the basis of the chronicity of the disease and its most serious and irreversible consequences [28]. These examples should lead to overturning the outdated paradigm of MCs as a mere effector of the acute phase. Mediators such as histamine and leukotrienes certainly cause the most dramatic effects of the MC-related response. However, MCs also orchestrate the late inflammation, communicating with other immune system cells such as eosinophils and fibroblasts, and mediating the chronicity of the inflammatory response and its long-term effects, as in CRSwNP and asthma.

Therefore, in the timeline of the immune response, MCs should not only appear in the acute phase but also in the maintenance and chronic phase.

This brings us to the second paradigm to be reconsidered. Historically, the drug that most notoriously influences the action of MCs is the antihistamine, which acts at the end of the activation of the cells, opposing its effects but not acting on its metabolism. This reinforces the concept that MCs are above all effector cells. Omalizumab also supports the same constraint, as it defuses the cells without intervening vastly in their function. However, a proof-of-concept trial on imatinib in severe asthma showed good results [130]. This demonstrates how acting on MCs upstream and considering MCs as a pathogenetic cornerstone of allergic and non-allergic inflammatory pathologies may design new therapeutic opportunities.

In conclusion, blocking mast cells is not just about blocking histamine. An integrated approach that considers MCs as a part of the immune response infrastructure as a whole could increase the response to therapy in pathologies where these rates are sometimes unsatisfactory, tempering both active and late chronic/fibrotic phases of the most disparate pathologies.

## Figures and Tables

**Figure 1 ijms-25-12615-f001:**
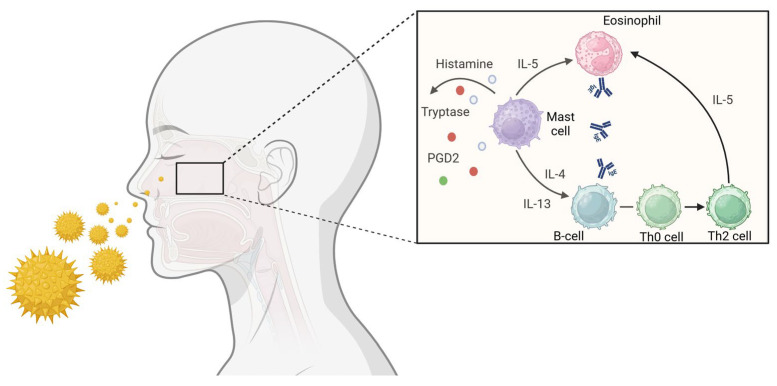
In the figure, it is shown how, when pollen comes into contact with the nasal mucosa of a person sensitized to this allergen, mast cells release histamine, tryptase, prostaglandins, and type 2 cytokines, including interleukin-4 (IL-4) and IL-13. Through degranulation, mast cells also indirectly promote the synthesis of IgE in B cells and contribute to the class switch of T cells from Th0 to Th2. Both Th2 lymphocytes and mast cells themselves are a source of IL-5, a molecule that amplifies the generation, survival, and activation of eosinophils. Ultimately, mast cells contribute to the shift from the Th0 to Th2 immune response. Created in BioRender.

**Figure 2 ijms-25-12615-f002:**
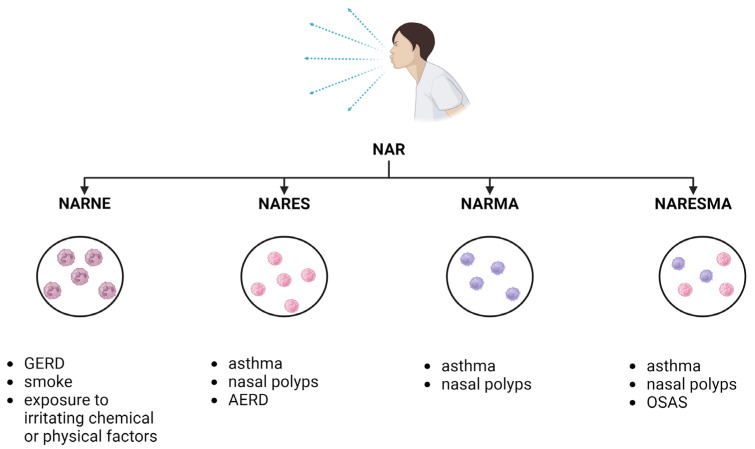
In the figure, the four main subgroups in non-allergic rhinitis (NAR) are shown with the respective main comorbidities: NARNE (NAR with neutrophils), NARES (NAR with eosinophilic syndrome), NARMA (NAR with mast cell), and NARESMA (NAR with eosinophils and mast cell). Acronyms: Gastroesophageal Reflux Disease (GERD); Aspirin-Exacerbated Respiratory Disease (AERD); Obstructive Sleep Apnea Syndrome (OSAS). Created in BioRender.

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
