# Peer review of "Mast Cells in Allergic and Non-Allergic Upper Airways Diseases: Sentinel in the Watchtower"

_ijms, 2024, doi:10.3390/ijms252312615_

Round 1

Reviewer 1 Report

Comments and Suggestions for Authors

In this review, Costanzo et al. provided a summary of the role of mast cells (MCs) in upper airway diseases and the treatments that focus on controlling MCs. The content could be beneficial for those studying upper airway diseases. However, this reviewer found some sentences that could have been more clearly articulated or may have overstated the role of MCs. These should be modified before publication.

1. Line 26-27: The statement "MCs originate in the bone marrow" is confusing as it contradicts the later description that some MCs also originate from the yolk sac. Please rephrase it.

2. Lines 61 and 72: "FcRI", gamma or epsilon should be included. 

3. Lines 104-106: "MCs contribute to the development and severity of T2 inflammatory diseases" is misleading as MCs are involved in disease severity regardless of T2-high or low, at least in asthma. 

Reference: Maun HR et al. 179, 417-431, 2019, Cell. 

4. Lines 172-173: MCs are not considered a primary inducer of B cell IgE class-switching or the cells that prime naïve T cells to differentiate into Th2 cells. It needs to be rephrased for clarity. 

5. Line 397: The abbreviation "INCS" should be explained at its first appearance. 

6. Line 447: It is unclear if dupilumab's therapeutic effect on CRSwNP is mediated by controlling MC activation. Evidence supporting this should be provided, along with references, or rephrase the sentence. 

7. Figure 2: "Hystamine" should be corrected to "Histamine," and the names of the cells in the figure should be described.

Author Response

Comments and Suggestions for Authors

In this review, Costanzo et al. provided a summary of the role of mast cells (MCs) in upper airway diseases and the treatments that focus on controlling MCs. The content could be beneficial for those studying upper airway diseases. However, this reviewer found some sentences that could have been more clearly articulated or may have overstated the role of MCs. These should be modified before publication.

Comment 1. Line 26-27: The statement "MCs originate in the bone marrow" is confusing as it contradicts the later description that some MCs also originate from the yolk sac. Please rephrase it.

Response 1: Thank you for pointing this out. We agree with this comment. Therefore, we have updated the sentences according to the suggestion [lines 26 and 29]. “In adults, MCs originate in the bone marrow from multipotent hematopoietic stem cells through a granulocyte/monocyte lineage, CD34+ and CD117+ […]. Fetal immature hypogranular MCs derived from stem cells can be found starting from the early stages of embryonic development in the yolk sac and the fetal liver

Comment 2. Lines 61 and 72: "FcRI", gamma or epsilon should be included.

Response 2: Thank you for pointing this out. We agree with this comment. Therefore, we have updated the sentences according to the suggestion [lines 55 and 66].

Comment 3. Lines 104-106: "MCs contribute to the development and severity of T2 inflammatory diseases" is misleading as MCs are involved in disease severity regardless of T2-high or low, at least in asthma. Reference: Maun HR et al. 179, 417-431, 2019, Cell.

Response 3: Thank you for pointing this out. We agree with this comment. Therefore, we have added a sentence according to the suggestion in lines 101-104, including the suggested reference. “However, it has been demonstrated that, at least in asthma, MCs contribute to the pathogenesis of the disease regardless of the T2 high or low phenotype, as mast cell tryptase is elevated in severe asthma patients independent of type 2 biomarker status [37].

Comment 4. Lines 172-173: MCs are not considered a primary inducer of B cell IgE class-switching or the cells that prime naïve T cells to differentiate into Th2 cells. It needs to be rephrased for clarity.

Response 4: Thank you for pointing this out. We agree with this comment. Therefore, we deleted the sentences.

Comment 5. Line 397: The abbreviation "INCS" should be explained at its first appearance.

Response 5: Thank you for pointing this out. We have updated the sentence according to the suggestion in line 406. “As advocated by 2016 Allergic Rhinitis and its Impact on Asthma (ARIA) guidelines, in patients with AR, the use of intranasal corticosteroid (INCS) is suggested as first-line therapy…”

Comment 6. Line 447: It is unclear if dupilumab's therapeutic effect on CRSwNP is mediated by controlling MC activation. Evidence supporting this should be provided, along with references, or rephrase the sentence.

Response 6: Thank you for pointing this out. We agree with this comment. Therefore, we rephrased the sentence underlining the indirect effect of dupilumab on MCs in regulating IgE production in lines 458-459. We also provided further references. [Zoabi Y, Levi-Schaffer F, Eliashar R. Allergic Rhinitis: Pathophysiology and Treatment Focusing on Mast Cells. Biomedicines. 2022 Oct 5;10(10):2486. doi: 10.3390/biomedicines10102486. PMID: 36289748; PMCID: PMC9599528.]. Dupilumab blocks the IL-4Rα subunit of IL-4 and IL-13 receptors, thus inhibiting IgE production and indirectly limiting MCs’ activation and degranulation.

Comment 7. Figure 2: "Hystamine" should be corrected to "Histamine," and the names of the cells in the figure should be described.

Response 7: Thank you for pointing this out. We corrected the figure.

Reviewer 2 Report

Comments and Suggestions for Authors

The review of Costanzo et al is a summary of information concerning mast cells in different upper airway diseases.

General comment 1: It should be made clearer in this review which novel and unique aspects this review adds to the field, as reviews, like references 34 and 35 of this review already summarize the role of mast cells in upper airways diseases.

General comment 2: I would suggest to make an overview table of all the upper airway diseases described in this review with the following columns: disease (list all mentioned diseases), mast cell numbers increased (yes/no) per disease, disease-specific roles of mast cells found, mast cell directed therapeutic trials/recommendations, efficacy of the respective therapies, literature references. This would give a good overview concerning the described topic in this review.

General comment 3: There is a lot written about the diseases, their definition and prevalence, but not so much about mast cells. The title however puts a strong focus on mast cells. I would therefore suggest another title, such as “mast cells in allergic and non-allergic upper airways diseases”, or something alike.

Specific comments:

#1 Line 41-44 and Figure 1. I think that the highly simplified depiction of MCT and MCTCs and their functions based on this distinction is outdated. By showing only MCT and MCTCs in Figure 1 with their suggested functions, too much emphasis is put on this old view, while mast cells of both types have been found to be involved in these functions. Recent work, also cited within these lines (ref. 12) shows that mast cells are more complex and that, for example, MCTs are involved in lung fibrosis. MCTCs do interact with immune cells by cytokine secretion (e.g. doi: 10.1007/s00403-008-0874-x and ). I would therefore suggest to omit or adapt figure 1.

#2 line 57: There is one well-known IgE receptor, whose activation leads to allergic response and that is FceRI. CD23 does not. Please use the singular form here.

#3: line 62: same as with FceRI: I would use the singular form of C5a receptor here.

#4 Figure 2:  This figure does not provide much information: Why is there a dandelion? Why not a classical allergen or something suggestive to trigger non-allergic rhinitis (e.g. a virus/bacteria/etc.)? The immune cells should be labelled, or a legend should be added. In the figure legend, it is written that mast cells cause a shift from Th0 to Th2, but this is not shown in the figure.

#5 Figure 3: It would make it more clear, if NAR would be written on top of the line with the arrows to show that it is an umbrella term for the shown endotypes.

#6 line 403: Histamine is an amine, not a protein. I highly and strongly recommend to the authors of this review on mast cells to once have a look at its structure to remember this forever.

Comments on the Quality of English Language

#1 line 72: Please insert an epsilon for the abbreviation. FceRI.

#2 line 137: please re-phrase: “Th2 cells, including mast cells themselves,” Mast cells are no Th2 cells, but in this sentence, it sounds like it.

#3 line 156-159: This sentence is quite cumbersome. Try to rephrase it and break it down to several sentences. And I would suggest in line 156: MCs are the main “cells” not “ones”.

#4 line 188: please use NAR at the first time, it is mentioned: in the title in line 188: 2.2. Non-allergic rhinitis (NAR) same for INCS: spell it out the first time it is mentioned in line 397 and not any more in line 434.

#5 line 294: I do not understand “a more relevant severe”. Please re-phrase.

#6 line 357: “synergize” might fit better here than “combine”.

Author Response

Response to Reviewer 2

Summary                          

Comments and Suggestions for Authors

The review of Costanzo et al is a summary of information concerning mast cells in different upper airway diseases.

General comment 1: It should be made clearer in this review which novel and unique aspects this review adds to the field, as reviews, like references 34 and 35 of this review already summarize the role of mast cells in upper airways diseases.

Response 1: Thank you for your precious comment. We added a paragraph describing the scope and the methodology used to write this review and how it may summarize many aspects of different diseases with a focus on mast cell, from a clinical to a therapeutical point of view [lines 112-123].

"In this review, we will describe the role of mast cells in the main allergic and non-allergic upper airway diseases. Proposing an in-depth description of the biological functions of mast cells is beyond the scope of this article. Therefore, only those aspects useful for understanding the pathogenesis of the diseases discussed below have been summarized. In the following sections, allergic rhinitis, chronic rhinosinusitis with nasal polyposis and non-allergic rhinitis will be addressed and analyzed in their general aspects and in such a way as to highlight aspects in common, with particular attention to the role of mast cells. In particular, non-allergic rhinitis will be addressed in its clinical and cytological features as we believe it is an often overlooked topic with important research potential. Finally, in light of the knowledge that has emerged on the functioning of the pathologies described, a section will be dedicated to management, with a particular focus on current and future therapeutic strategies that target mast cells."

General comment 2: I would suggest to make an overview table of all the upper airway diseases described in this review with the following columns: disease (list all mentioned diseases), mast cell numbers increased (yes/no) per disease, disease-specific roles of mast cells found, mast cell directed therapeutic trials/recommendations, efficacy of the respective therapies, literature references. This would give a good overview concerning the described topic in this review.

Response 2: Thank you for your valuable comment. Some issues arose during the compilation of the suggested table. Regarding the number of mast cells in the various pathologies, there is little relevance on the number of these cells in allergic rhinitis as the biopsy is rarely performed, while the cytology data are already reported in the section dedicated to non-allergic rhinitis; regarding chronic rhinosinusitis with nasal polyps, the disease-specific roles of mast cells are the same as those reported in the remaining airway pathologies, i.e. an indication of a more serious disease. Regarding the mast cell-directed therapeutic trials/recommendations and efficacy, the treatment of the three pathologies is often overlapped (intranasal corticosteroids and antihistamines). The use of biological drugs is indicated only in severe rhinosinusitis with nasal polyposis and the data on efficacy and effects on mast cells are already reported in the text. For all the reasons listed above, although we appreciate and agree with the suggestion to add a table, in this specific case we believe that including this element may be redundant.

General comment 3: There is a lot written about the diseases, their definition and prevalence, but not so much about mast cells. The title however puts a strong focus on mast cells. I would therefore suggest another title, such as “mast cells in allergic and non-allergic upper airways diseases”, or something alike.

Response 3: Thank you for pointing this out. We agree with your comment. Therefore, we update the title according to the suggestion. “Mast Cells in Allergic and Non-Allergic Upper Airways Diseases: Sentinel in the Watchtower”.

Specific comments

#1 Line 41-44 and Figure 1. I think that the highly simplified depiction of MCT and MCTCs and their functions based on this distinction is outdated. By showing only MCT and MCTCs in Figure 1 with their suggested functions, too much emphasis is put on this old view, while mast cells of both types have been found to be involved in these functions. Recent work, also cited within these lines (ref. 12) shows that mast cells are more complex and that, for example, MCTs are involved in lung fibrosis. MCTCs do interact with immune cells by cytokine secretion (e.g. doi: 10.1007/s00403-008-0874-x and ). I would therefore suggest to omit or adapt figure 1.

Response 1: Thank you for pointing this out. We agree with this comment. Therefore, we deleted the figure.

#2 line 57: There is one well-known IgE receptor, whose activation leads to allergic response and that is FceRI. CD23 does not. Please use the singular form here.

Response 2: Thank you for pointing this out. We have updated the sentence according to the suggestion [line 50]. "The most well-known is IgE receptor, whose activation leads to allergic response".

#3: line 62: same as with FceRI: I would use the singular form of C5a receptor here.

Response 3: Thank you for pointing this out. We have updated the sentence according to the suggestion [line 56]. "Among the non-IgE receptors, we can distinguish FcγRI receptors that allow mast cells to bind to IgG, Toll-like receptors, C5a receptors."

#4 Figure 2:  This figure does not provide much information: Why is there a dandelion? Why not a classical allergen or something suggestive to trigger non-allergic rhinitis (e.g. a virus/bacteria/etc.)? The immune cells should be labelled, or a legend should be added. In the figure legend, it is written that mast cells cause a shift from Th0 to Th2, but this is not shown in the figure.

Response 4: Thank you for pointing this out. We have updated the figure according to the suggestions and omitting the description the last sentence. "In the figure, it is shown how, when pollen comes into contact with the nasal mucosa of a person sensitized to this allergen, mast cells release histamine, tryptase, prostaglandins, and type 2 cytokines, including interleukin-4 IL-4 and IL-13. Through degranulation, mast cells also promote the synthesis of IgE in B cells, amplify the generation and survival of eosinophils, and initiate the shift from Th0 to Th2 immune response."

#5 Figure 3: It would make it more clear, if NAR would be written on top of the line with the arrows to show that it is an umbrella term for the shown endotypes.

Response 5: Thank you for pointing this out. We have updated the figure according to the suggestions.

#6 line 403: Histamine is an amine, not a protein. I highly and strongly recommend to the authors of this review on mast cells to once have a look at its structure to remember this forever.

Response 6: Thank you for pointing this out. We have corrected the error in the sentence where it was present [line 411]. “Antihistamines, both oral and local, limit the effect of histamine, the main protein mediator MC produces and the principal responsible for most symptoms such as nose itching”.

Comments on the Quality of English Language

#1 line 72: Please insert an epsilon for the abbreviation. FceRI.

Response 1: Thank you for pointing this out. We agree with this comment. Therefore, we have updated the sentences according to the suggestion [line 66].

#2 line 137: please re-phrase: “Th2 cells, including mast cells themselves,” Mast cells are no Th2 cells, but in this sentence, it sounds like it.

Response 2: Thank you for pointing this out. We agree with this comment. Therefore, we have updated the sentences according to the suggestion [line 147]. “In particular, it was found that histamine produced from MCs’ degranulation initiates the loss of the epithelial barrier during the early stage of the allergic immune response, which is then maintained by Th2 cells and including MCs themself, through the production of interleukin-4 (IL-4) and IL-13

#3 line 156-159: This sentence is quite cumbersome. Try to rephrase it and break it down to several sentences. And I would suggest in line 156: MCs are the main “cells” not “ones”.

Response 3: Thank you for pointing this out. We agree with this comment. Therefore, we have updated the sentences according to the suggestion [line 166]. "In AR, MCs are the main cells responsible for the symptoms and a key element in every stage of the allergy. Indeed, in the early stage MCs are the first cells to be activated and triggered determining the immediate reaction. In the late inflammatory phase, they organize the cellular recruitment that perpetuates clinical chronicity"

#4 line 188: please use NAR at the first time, it is mentioned: in the title in line 188: 2.2. Non-allergic rhinitis (NAR) same for INCS: spell it out the first time it is mentioned in line 397 and not any more in line 434.

Response 4: Thank you for pointing this out. We have updated the sentences according to the suggestions [lines 196, 405 and 442]

#5 line 294: I do not understand “a more relevant severe”. Please re-phrase.

Response 5: Thank you for pointing this out. We agree with this comment. Therefore, we have corrected the typo according to the suggestion [line 301]. “In particular, the presence of MCs on nasal cytology indicates a more relevant severe and does not represent mere descriptive data, but an important element characterizing NAR.”

#6 line 357: “synergize” might fit better here than “combine”.

Response 6: Thank you for pointing this out. We agree with this comment. Therefore, we have updated the sentence according to the suggestion [line 365].

Round 2

Reviewer 2 Report

Comments and Suggestions for Authors

Thank you for changing your manuscript accordingly. I have two more comments:

@ response 4: Figure 2: The figure improved a lot already. If you show IL-5 and T cells, you should also describe them in your figure legend. I didn't mean to delete the part that Th0 shifts to Th2, but that you should show this in your figure. Just add another T cell. Call the first one Th0 and the one furhter to the right Th2 and let the arrow of IL-5 go from the Th2 cell to the eosinophil and leave the part about the Thelper cell shift in the figure legend. 

@ Response 5: line 103: I would add the word "condition" or "phenotype" or something alike: "In particular, the presence of MCs on nasal cytology indicates a more severe [condition/phenotype] and does not represent mere descriptive data, 302 but an important element characterizing NAR." or "higher/increased severity" instead of "a more severe"

Author Response

Response to Reviewer 2_Round 2

Summary                          

Thank you very much for taking the time to review this manuscript again and for the precious suggestions which improved the quality of the article overall. Please find the detailed responses below and the corresponding revisions/corrections highlighted/in track changes in the re-submitted files.

Comments:

Thank you for changing your manuscript accordingly. I have two more comments:

@ response 4: Figure 2: The figure improved a lot already. If you show IL-5 and T cells, you should also describe them in your figure legend. I didn't mean to delete the part that Th0 shifts to Th2, but that you should show this in your figure. Just add another T cell. Call the first one Th0 and the one furhter to the right Th2 and let the arrow of IL-5 go from the Th2 cell to the eosinophil and leave the part about the Thelper cell shift in the figure legend.

Response 4: Thank you for pointing this out. We have updated the figure and legend according to the suggestion.

"In the figure, it is shown how, when pollen comes into contact with the nasal mucosa of a person sensitized to this allergen, mast cells release histamine, tryptase, prostaglandins, and type 2 cytokines, including interleukin-4 (IL-4) and IL-13. Through degranulation, mast cells also indirectly promote the synthesis of IgE in B cells and contribute to the class switch of T cells from Th0 to Th2. Both Th2 lymphocytes and mast cells themselves are a source of IL-5, a molecule that amplifies the generation, survival, and activation of eosinophils. Ultimately, mast cells contribute to the shift from the Th0 to Th2 immune response. Created in BioRender"

@ Response 5: line 103: I would add the word "condition" or "phenotype" or something alike: "In particular, the presence of MCs on nasal cytology indicates a more severe [condition/phenotype] and does not represent mere descriptive data, 302 but an important element characterizing NAR." or "higher/increased severity" instead of "a more severe"

Response 5: Thank you for pointing this out. We have updated the sentence according to the suggestion [line 302]. "In particular, the presence of MCs on nasal cytology indicates a more severe condition and does not represent mere descriptive data, but an important element characterizing NAR."
